# Rotational Distortion and Compensation in Optical Coherence Tomography with Anisotropic Pixel Resolution

**DOI:** 10.3390/bioengineering10030313

**Published:** 2023-03-01

**Authors:** Guangying Ma, Taeyoon Son, Tobiloba Adejumo, Xincheng Yao

**Affiliations:** 1Department of Biomedical Engineering, University of Illinois Chicago, Chicago, IL 60607, USA; 2Department of Ophthalmology and Visual Sciences, University of Illinois Chicago, Chicago, IL 60612, USA

**Keywords:** eye movement, image registration, optical coherence tomography, optical coherence tomography angiography

## Abstract

Accurate image registration is essential for eye movement compensation in optical coherence tomography (OCT) and OCT angiography (OCTA). The spatial resolution of an OCT instrument is typically anisotropic, i.e., has different resolutions in the lateral and axial dimensions. When OCT images have anisotropic pixel resolution, residual distortion (RD) and false translation (FT) are always observed after image registration for rotational movement. In this study, RD and FT were quantitively analyzed over different degrees of rotational movement and various lateral and axial pixel resolution ratio ($R_{L}/R_{A}$) values. The RD and FT provide the evaluation criteria for image registration. The theoretical analysis confirmed that the RD and FT increase significantly with the rotation degree and $R_{L}/R_{A}$. An image resizing assisting registration (RAR) strategy was proposed for accurate image registration. The performance of direct registration (DR) and RAR for retinal OCT and OCTA images were quantitatively compared. Experimental results confirmed that unnormalized $R_{L}/R_{A}$ causes RD and FT; RAR can effectively improve the performance of OCT and OCTA image registration and distortion compensation.

## 1. Introduction

Optical coherence tomography (OCT) is a non-invasive medical imaging technology that has been broadly used in ophthalmology clinics and research laboratories [1,2,3]. For clinical OCT systems, the acquisition of a 3D volume normally takes several seconds. Within the acquisition, a fixation target is usually used to minimize voluntary eye movements. However, there are still inevitable involuntary eye movements [4], which cause image drift and distortion that affect the image quality and clinical interpretation. Different approaches have been introduced to compensate for eye movements [5]. Image registration is a simple and low-cost strategy widely used in commercial systems [6,7,8]. Image registration benefits OCT data processing from two aspects. First, image registration can help to align repeated B-scans at the same location. OCT images inherently suffer from speckle noise. Averaging a few images taken from the same location is a common way to reduce noise [9,10,11,12,13]. However, averaging is always affected by the image displacement caused by eye movement. Therefore, image registration is frequently used to compensate for the movement to enhance image quality. Additionally, registered repeated B-scans are the prerequisites for OCT angiography (OCTA) data processing algorithms [14,15]. Established OCTA processing methods, such as speckle-variance [16], phase-variance [17], optical microangiography (OMAG) [18], and split-spectrum amplitude-decorrelation angiography (SSADA) [19], compute the signal variance of a certain number of repeated B-scans. With the assumption that the signal of blood flow is variable compared to that of static tissues, the blood vasculature can be extracted by computing the signal variance of each pixel [16,20,21]. As the computation assumes that the B-scans are recorded exactly at the same location, the repeated images have to be registered precisely to compensate for the movement and allow pixel-wise comparison. Second, registration between the consecutive B-scans is required to correct the vasculature distortion and discontinuity in en face OCTA images [4,22].

Image registration is also an imperative step for recently emerging optoretinography (ORG) [23]. ORG measures the stimulus-evoked intrinsic optical signal (IOS) changes, which reflect the functional status of retinal photoreceptors and inner neurons [23]. As ORG has a much higher spatial resolution than electroretinography (ERG), it has been actively studied in recent years [24,25,26,27,28,29,30,31]. OCT-based ORG maps the IOS amplitude [25,30] and phase [26,27,28,31] changes in the retina evoked by visible light stimulation. An ORG study typically records the retina in three phases including pre-stimulation, stimulation on, and post-stimulation. Therefore, a relatively long imaging time is needed, and inevitable eye movements may occur during the image acquisition. On the other hand, the photoreceptor outer segment shrinkage, a typical phenomenon observed in ORG, is only several hundred nanometers [26,29]. A small image movement will greatly affect the interpretation reliability of the experimental result.

OCT image registration has been studied since it started to be applied in ophthalmology [2,5]. The common strategy is to register the image displacement by applying a transformation matrix to A-lines or B-scans. The transformation matrix is optimized by maximizing the cross-correlation coefficient between the images to be registered. In previous studies, only x, y, and z direction shifts were considered [22,32,33,34]. However, the rotational movement always happens simultaneously with translational movement [4]. Therefore, for precise registration, compensation for the rotational movement should also be considered.

In OCT, it is known that the lateral resolution is diffraction-limited, while the axial resolution is defined by the spectral bandwidth of the light source [1]. The OCT instruments typically have anisotropic resolutions in the lateral and axial dimensions. Recently, we tried to compensate for eye movement by registering the B-scans for both rotational and translational degrees of freedom. We observed that residual distortion (RD) and false translation (FT) always existed in the OCT images with anisotropic pixel resolution even after the registration. The *RD* and *FT* provide the evaluation criteria for image registration.

In the following sections, we will first demonstrate a conceptual simulation of the effect of the lateral and axial resolution ratio ($R_{L}/R_{A}$) on rotational distortion, and then experimentally validate an image resizing assisting registration (RAR) approach for rotational distortion compensation in OCT and OCTA images. Both theoretical analysis and experimental results confirmed that $R_{L}/R_{A}$ normalization can significantly improve registration performance. 

## 2. Theoretical Simulation of Rotational Distortion and Compensation

### 2.1. Conceptual Illustration of Rotational Distortion

Figure 1 is a schematic illustration of the compensation performance of rotational movements in OCT images with isotropic ($R_{L}/R_{A}$ = 1) and anisotropic ($R_{L}/R_{A}$ ≠ 1) resolutions. Figure 1a shows the first (1st) scan with OCT illumination perpendicular to the bottom line of the triangular target, and Figure 1b corresponds to the second (2nd) scan of the same target with a rotational movement. As shown in Figure 1c–e, the rotational displacement in the OCT with isotropic resolution ($R_{L}/R_{A}$ = 1) can be perfectly corrected. On the contrary, the image distortion remained in the registered scan with anisotropic resolution ($R_{L}/R_{A}$ ≠ 1) after the rotational correction (Figure 1f–h). 

### 2.2. Quantitative Simulation of Rotational Movement

Figure 2 illustrates the procedures for quantitative assessment of the rotational eye movement and image registration performance. Figure 2a is a simulated retinal B-scan, working as the reference frame, without rotational movement. Each pixel signal can be represented as $P_{s}\left(x_{s},y_{s}\right)$ in Cartesian coordinates and $P_{s}\left(r_{s},\theta_{s}\right)$ in polar coordinates, which follows the relationship between Cartesian and polar coordinate systems (1).
(1)$$\begin{array}{c} x=r\cos\left(\theta \right),\;y=r\sin\left(\theta \right)\\ r=\sqrt{x^{2}+y^{2}},\;\theta =\tan^{-1}\left(\frac{y}{x}\right) \end{array}$$

Figure 2b shows angular rotation caused by an eye movement. As our focus is the rotational movement, to simplify the simulation, the translation movement is assumed as 0, and the rotational movement is denoted as ϕ. As the retinal position is typically guided by the fixation target in OCT, the pivot is assumed at the center of the fovea, noted as $P_{r}\left(0,0\right)$ in Figure 2b. To simplify the calculation, the origin of the coordinate system is set at the pivot. The corresponding image pixel with rotational movement is represented as $P_{r}\left(x_{r},y_{r}\right)$ in Cartesian coordinates and $P_{r}\left(r_{r},\theta_{r}\right)$ in polar coordinates.

Considering the rotational movement only, the relationship between the source frame $P_{s}$ and rotated frame $P_{r}$ is shown as follows.
(2)$$P_{r}\left(r_{r},\theta_{r}\right)=P_{s}\left(r_{s},(\theta_{s}+\phi )\right)$$

Therefore, the transformation of a certain point in $P_{s}$ to $P_{r}$ can be derived as (3) in polar coordinates and (4) in Cartesian coordinates.
(3)$$\begin{array}{c} r_{r}=r_{s}=\sqrt{x_{s}^{2}+y_{s}^{2}}\\ \theta_{r}=(\theta_{s}+\phi )=\tan^{-1}\left(\frac{y_{s}}{x_{s}}\right)+\phi \end{array}$$
(4)$$\begin{array}{c} x_{r}=\sqrt{x_{s}^{2}+y_{s}^{2}}\cos\left(\tan^{-1}\left(\frac{y_{s}}{x_{s}}\right)+\phi \right)\\ y_{r}=\sqrt{x_{s}^{2}+y_{s}^{2}}\sin\left(\tan^{-1}\left(\frac{y_{s}}{x_{s}}\right)+\phi \right) \end{array}$$

To evaluate the effect of the $R_{L}/R_{A}$, the source frame and rotated frame can be resized as follows: (5)$$x_{s}^{\prime }=\frac{x_{s}}{n},\;\;y_{s}^{\prime }=y_{s}$$
(6)$$x_{r}^{\prime }=\frac{x_{r}}{n},\;\;y_{r}^{\prime }=y_{r}$$
where n = $R_{L}/R_{A}$. Thus, the resized $P_{s}\left(x_{s},y_{s}\right)$ and $P_{r}\left(x_{r},y_{r}\right)$ can be represented as $P_{s}^{\prime }\left(x_{s}^{\prime },y_{s}^{\prime }\right)$ (Figure 2c) and $P_{r}^{\prime }\left(x_{r}^{\prime },y_{r}^{\prime }\right)$ (Figure 2d). Then, Equation (7) can be derived from (4)–(6).
(7)$$\begin{array}{c} x_{r}^{\prime }=\frac{1}{n}\sqrt{{\left(nx_{s}^{\prime }\right)}^{2}+y_{s}^{\prime }{}^{2}}\cos\left(\tan^{-1}\left(\frac{y_{s}^{\prime }}{nx_{s}{}^{\prime }}\right)+\phi \right)\\ y_{r}^{\prime }=\frac{1}{n}\sqrt{{\left(nx_{s}^{\prime }\right)}^{2}+y_{s}^{\prime }{}^{2}}\sin\left(\tan^{-1}\left(\frac{y_{s}^{\prime }}{nx_{s}{}^{\prime }}\right)+\phi \right) \end{array}$$

Equation (7) represents the relationship between $P_{s}^{\prime }\left(x_{s}^{\prime },y_{s}^{\prime }\right)$ and $P_{r}^{\prime }\left(x_{r}^{\prime },y_{r}^{\prime }\right)$, which was used for further registration computation. 

### 2.3. Image Registration

Image $P_{r}^{\prime }$ was registered to $P_{s}^{\prime }$ by executing a rotation with an angle of $\phi^{\prime }$ followed by a translation with a vector of $t\left(x_{t},y_{t}\right)$. The registered image was denoted as $P_{r}^{\prime \prime }$. Therefore, the relationship between $P_{r}^{\prime \prime }$ and $P_{r}^{\prime }$ can be expressed as follows.
(8)$$P_{r}^{\prime \prime }{}_{step1}\left(r,\theta \right)=P_{r}^{\prime }\left(r,\;\left(\theta +\phi^{\prime }\right)\right)$$
(9)$$P_{r}^{\prime \prime }=P_{r}^{\prime \prime }{}_{step2}\left(x,y\right)=P_{r}^{\prime \prime }{}_{step1}\left(x,\;y\right)+t\left(x_{t},y_{t}\right)$$

The registration performance can be validated by
(10)$$D_{sum}\left(t\left(x_{t},y_{t}\right),\phi^{\prime }\right)=\sum \sum \sqrt{{\left(x_{s}^{\prime }{}_{ij}-x_{r}^{\prime \prime }{}_{ij}\right)}^{2}+{\left(y_{s}^{\prime }{}_{ij}-y_{r}^{\prime \prime }{}_{ij}\right)}^{2}}$$
where $P_{s}^{\prime }{}_{ij}\left(x_{s}^{\prime }{}_{ij},y_{s}^{\prime }{}_{ij}\right)$ and $P_{r}^{\prime \prime }{}_{ij}\left(x_{r}^{\prime \prime }{}_{ij},y_{r}^{\prime \prime }{}_{ij}\right)$ represent the same locations of the source image $P_{s}^{\prime }$ and the registered image $P_{r}^{\prime \prime }$. Therefore, $D_{sum}$ represents the sum of the on-image distance between $P_{s}^{\prime }{}_{ij}$ and $P_{r}^{\prime \prime }{}_{ij}$ with the unit of pixel. The optimized $\phi^{\prime }$ and t are achieved by minimizing $D_{sum}$. 

### 2.4. Displacement Characterization

The images are considered as properly registered when $\phi^{\prime }$ and $t\left(x_{t},y_{t}\right)$ are optimized (Figure 2f). Then, the RD is characterized. RD is defined as the distance between the location of an individual pixel between the resized source frame ($P_{s}^{\prime }\left(x_{s}^{\prime }{}_{ij},y_{s}^{\prime }{}_{ij}\right)$) and the registered rotated frame ($P_{r}^{\prime \prime }\left(x_{r}^{\prime \prime }{}_{ij},y_{r}^{\prime \prime }{}_{ij}\right)$), which is shown as (11),
(11)$$RD\left(i,j\right)=\sqrt{{\left(nx_{s}^{\prime }{}_{ij}-nx_{r}^{\prime \prime }{}_{ij}\right)}^{2}+{\left(y_{s}^{\prime }{}_{ij}-y_{r}^{\prime \prime }{}_{ij}\right)}^{2}}\;$$
where, RD(i,j) denotes the RD at lateral (i) and axial (j) location. The purpose of defining RD in the real dimension is to allow the comparison of RD over images of different $R_{L}/R_{A}$ values.

We assessed the registration performance over various ϕ and $R_{L}/R_{A}$ to investigate the relationship among ϕ, $R_{L}/R_{A}$, and RD. Corresponding to different $R_{L}/R_{A}$ and ϕ, the RDs are represented in Figure 3. It was observed that there is always RD after the registration of rotational movement for OCT images with anisotropic pixel resolution; RD increases significantly as $R_{L}/R_{A}$ and ϕ increase. The dark bands (relatively lower intensity) in the center of each image were because of the pivot (Figure 3). The bands were not symmetric when the ϕ was 2.5 or 1 degree, because the initial rotational movement we simulated was counterclockwise. 

### 2.5. Characterization of the Transformation Occurred during Registration

When the optimal registration is achieved, the corresponding transformation information is recorded (Figure 4). The rotational compensation ($\phi^{\prime }$) is represented in Figure 4a. The result showed that all $\phi^{\prime }$ increases with $R_{L}/R_{A}$ value increase; $\phi^{\prime }$ equals the initially introduced angle (ϕ) for $R_{L}/R_{A}$ = 1, whereas $\phi^{\prime }$ is larger than ϕ for $R_{L}/R_{A}$ ≠ 1. The translational compensation $t\left(x_{t},y_{t}\right)$ is represented in Figure 4b,c. Because only the rotational movement is introduced, the $t\left(x_{t},y_{t}\right)$ represents the translation artifact away from the original location. Thus, we can consider $t\left(x_{t},y_{t}\right)$ as FT. The result shows that FT increases as $R_{L}/R_{A}$ and ϕ angle increase. The lateral FT is significantly larger than the axial FT. Figure 4b,c indicates that if $R_{L}/R_{A}$ is not normalized, the registration for rotational movement will introduce FT to the images, especially in the lateral direction. We will describe the effects of FT on 3D OCTA registration in Section 4.2: Registration of 3D OCT image.

An explicit analysis can help us to understand how the $\phi^{\prime }$ and $t\left(x_{t},y_{t}\right)$ are affected by $R_{L}/R_{A}$. Figure 5a shows the reference frame $P_{s}$ and the rotated frame $P_{r}$, both with $R_{L}/R_{A}$ = 1. $P_{r}\left(x_{r},y_{r}\right)$ is matched with $P_{s}\left(x_{s},y_{s}\right)$ at the same location after image rotation by angle ϕ. Figure 5b shows $P_{s}^{\prime }\left(x_{s}^{\prime },y_{s}^{\prime }\right)$ and $P_{r}^{\prime }\left(x_{r}^{\prime },y_{r}^{\prime }\right)$ with $R_{L}/R_{A}$ = n. The original rotational angle ϕ is changed to ψ and defined as (12).
(12)$$\psi =\theta_{s}^{\prime }-\theta_{r}^{\prime }$$

ψ reflects how the ϕ is changed when $R_{L}/R_{A}$ is changed to n. We can obtain (13) by first converting the angle to triangular function and then substituting (5) and (6) for (12).
(13)$$\psi =\tan^{-1}\left(\frac{y_{s}^{\prime }}{x_{s}^{\prime }}\;\right)-\tan^{-1}\left(\frac{y_{r}^{\prime }}{x_{r}^{\prime }}\right)=\tan^{-1}\left(\frac{y_{s}}{\frac{x_{s}}{n}}\right)-\tan^{-1}\left(\frac{y_{r}}{\frac{x_{r}}{n}}\right)=\tan^{-1}\left(\frac{ny_{s}}{x_{s}}\right)-\tan^{-1}\left(\frac{ny_{r}}{x_{r}}\right)$$

As the retina is an elongated structure, for small $R_{L}/R_{A}$ values, the majority of the points satisfy the small angle approximation. Thus, (13) can be derived as (14).
(14)$$\psi =\left(\frac{ny_{s}}{x_{s}}\right)-\left(\frac{ny_{r}}{x_{r}}\right)=n\left(\frac{y_{s}}{x_{s}}-\frac{y_{r}}{x_{r}}\right)=n(\theta_{s}-\theta_{r})$$

Considering the definition of ϕ, the relationship between ψ and ϕ can be described as (15).
(15)ψ=nϕ

The ratio between ψ and ϕ (ψ/ϕ) equals n. Note that ψ corresponds to each pixel, and it is a function of location. (15) is correct only if the small angle approximation is satisfied. For large $R_{L}/R_{A}$ values, the image will be more compressed along the lateral direction, which causes the small angle approximation unsatisfied, and ψ/ϕ becomes smaller than n. 

As ψ reflects the ϕ after $R_{L}/R_{A}$ is changed to n, it can be considered as an approximation of $\phi^{\prime }$. Figure 5d depicts the ratio between $\phi^{\prime }$ and ϕ; it is close to n for small $R_{L}/R_{A}$ values and becoming smaller than n for the large $R_{L}/R_{A}$ values. This result supports that ψ is an approximation of $\phi^{\prime }$.

Figure 5c showed that if the rotation of ψ is compensated, $P_{r}^{\prime }$ is changed to $P_{r}^{\prime \prime }{}_{step1}$, $P_{r}^{\prime \prime }{}_{step1}\left(\theta_{r}^{\prime \prime }{}_{step1},r_{r}^{\prime \prime }{}_{step1}\right)$ is close to $P_{s}^{\prime }\left(\theta_{s}^{\prime },r_{s}^{\prime }\right)$. However, they are not completely matched, because after the rotation of $P_{r}^{\prime \prime }{}_{step1}$ by ψ, $\theta_{r}^{\prime \prime }{}_{step1}=\theta_{s}^{\prime }$, whereas $r_{r}^{\prime \prime }{}_{step1}<r_{s}^{\prime }$. It was already demonstrated in the illustration (Figure 5c) that after the rotation, there was still a residual between the two points. The lateral residual was proportional to sine($\theta_{s}^{\prime }$), and the axial residual was proportional to cosine($\theta_{s}^{\prime }$). For the second step, these residuals would be compensated; therefore, the compensated lateral residual is much larger than the axial residual. 

Note that the real registration computes the two steps simultaneously, then assesses the final result. Thus, in the optimized registration, it is not necessary to rotate exact ψ followed by a $t\left(x_{t},y_{t}\right)$. It can be a rotation close to ψ with a corresponding $t\left(x_{t},y_{t}\right)$ together, which gives better $D_{sum}$. Therefore, the real computation result can be different from this explicit analysis. Nevertheless, this analysis helps to understand the relationship between ϕ, $\phi^{\prime }$, and $t\left(x_{t},y_{t}\right)$, especially for the scenario when $R_{L}/R_{A}$ is not too large.

### 2.6. Summary

In summary, the theoretical simulation indicates that for OCT image registration, there is always RD and FT if the $R_{L}/R_{A}$ ≠ 1. A preprocessing procedure to normalize the image to $R_{L}/R_{A}$ = 1 is important for accurate registration to correct rotational movement.

## 3. Materials and Methods for Experimental Validation

### 3.1. Human Subjects

This study was approved by the Institutional Review Board of the University of Illinois at Chicago and followed the ethical standards stated in the Declaration of Helsinki. Each subject provided informed consent before participation in the research. The repeated OCT B-scans and the 3D OCT/OCTA images were acquired from a healthy 27-year-old female and a healthy 36-year-old male, respectively.

### 3.2. Imaging System and Data Acquisition

A custom-designed SD-OCT was developed for human retina imaging (Figure 6). Briefly, a broadband (M-T-850-HP-I, Superlum, Cork, Ireland, $\lambda_{center}=850\;\mathrm{nm},\;\Delta \lambda =165\;\mathrm{nm}$) superluminescent diode (SLD) was used as the light source. A fiber coupler (TW850R2A1, Thorlabs, Newton, NJ, USA; 90:10) divided the OCT light to the sample (10%) and reference (90%) arms. A custom-designed spectrometer was constructed with a line CCD camera (AViiVA EM4, E2V Technologies, Chelmsford, UK; 2048 pixels) and a transmission grating (Wasatch Photonics, West Logan, UT, USA; 1200 line/mm). The axial and lateral pixel resolutions were 1 µm and 10 µm, respectively. A fixation target with red light was used to minimize voluntary eye movements. For OCT recording, the illumination power on the cornea was ∼600 μW. The repeated OCT B-scans were acquired from the macular region covering 5.4 mm retina, ~18 eccentricity degrees. The total B-scan number was 90 with a frame speed of 100 frames per second (fps). The 3D OCT volume was acquired at the macular region with a size of 3.5 mm × 3.5 mm for the OCTA image. The number of B-scan repetitions for OCTA construction was 4. The frame speed was 200 fps. The total recording time was 7 s.

### 3.3. Data Processing

OCT B-scans were reconstructed from raw data through k-sampling, numerical dispersion compensation, Hanning windowing, fixed pattern removal, and fast Fourier transformation (FFT). Then, RAR was performed following the workflow of resizing the image to $R_{L}/R_{A}$ = 1, registration, and resizing the image to the original $R_{L}/R_{A}$ (Figure 7). The image was resized using the Matlab (R2021a, Portola Valley, CA, USA) built-in function “imresize”, with interpolation mode “nearest”. The images were registered by the ImageJ plugin “MultiStackReg” (v1.4) with mode “rigid”. “MultiStackReg” is based on the minimization of the mean square intensity difference between a reference and a moving image [5,35], which has been demonstrated to be effective for retinal OCT images [12,13,22,36]. Choosing the “rigid” mode was to compensate for both translational and rotational eye movement. The compensated rotational and translational movements (Figure 8a,b) were calculated from the transformation matrix file generated by “MultiStackReg”. The 2D correlation coefficient (CC) of repeated B-scan data (Figure 8b) was computed by Matlab built-in function “corr2”, and a single CC point was computed using each of the images and the mean image of the stack. The CC of the consecutive B-scans of the 3D OCT/OCTA data was computed in two steps: first, to obtain the mean B-scan by averaging the B-scans at the same location (every 4 frames), and second, to calculate the CC of the mean B-scans at the adjacent locations. The image line profile was computed by the ImageJ function “Plot profile” with the line width “5”.

The OCTA signal was computed via the speckle variance method; in other words, the hemodynamic signal was extracted by computing the intensity variance of the repeated B-scans. For post-processing, contrast enhancement and strip suppression were used to enhance the hemodynamic signal. 

## 4. Experimental Results

### 4.1. Registration of Repeated B-Scans

Without registration, the retinal repeated OCT B-scans were unstable (Appendix A) because of eye movement. To compensate for the movement, direct registration (DR) and RAR were applied to the image stack. The result showed that both DR and RAR can stabilize the images. However, the image stack with DR was not perfectly registered, and the image distortion was observed over frames (Figure 8(c1) and Appendix A); with RAR, the image stack was well overlapped, and no significant movement was observed (Figure 8(c2) and Appendix A).

The rotational and translational compensation were recorded and plotted in Figure 8a. The rotational compensation of RAR was smaller than DR. The lateral translational compensation of RAR and DR was different, whereas the axial translational compensation was very similar. These results were consistent with the theoretical simulation (Figure 4). The registration performance was validated by the 2D correlation coefficient (CC) (Figure 8b). The CC of RAR was higher than DR for every frame, which means that RAR was more effective than DR. The registration performance was further investigated by comparing the mean image of the entire stack. The mean image of DR (Figure 8(c1)) was more blurred than that of RAR (Figure 8(c2)). We evaluated the sharpness of the images by the gradient method [37]. The sharpness of the mean image of RAR was 15% higher than that of DR. The detailed differences were validated by the zoom-in view and line profiles (Figure 8d). The mean image of RAR showed a clearer structure and better separation of individual retinal bands. For instance, Bruch’s membrane was more visible in the mean image of RAR.

### 4.2. Registration of 3D OCT Image

Figure 9a shows the resliced B-scan along the slow scan direction of OCT 3D data. Before the registration, severe eye movement was observed (Figure 9(a1)). After DR (Figure 9(a2)) and RAR (Figure 9(a3)), the distortion caused by the eye movement was significantly reduced. The registration information is recorded and plotted in Figure 9b. Similarly to Figure 8a, the rotational compensation of RAR was smaller than that of DR, the lateral translation of RAR was different from that of DR, and the axial translation of RAR was very similar to that of DR. These results are consistent with those of the theoretical simulation (Figure 4). The registration performance is validated in Figure 9c. It shows the CC of two frames at two consecutive locations. For each of the consecutive B-scan pairs, the CC of RAR is larger than DR. 

Figure 10a shows representative en face OCTA processed with DR and RAR. The black regions at the top and bottom areas were caused by zero filling when part of the image moved out of the boundary during the image registration. The zoom-in view of the OCTA en face image with RAR showed better vessel connectivity (Figure 10(b1)), less vessel distortion (Figure 10(b2), and a smoother vessel boundary (Figure 10(b3)). Additionally, better capillary vessel visibility was observed in all of the zoom-in views of RAR. Figure 10c shows the representative OCTA B-scans processed by DR and RAR from the same location. It is observed that the OCTA B-scans of DR (Figure 10(c1)) have a higher background signal than that of RAR (Figure 10(c2)). Figure 10d shows the representative OCT B-scans of the same frame registered by DR and RAR. The black region was caused by zero filling during the registration. This black region was smaller in RAR because the rotational and axial translational compensation of RAR was smaller than that of DR.

## 5. Discussion

To the best of our knowledge, this is the first comprehensive study of the compensation for the rotational movement of retinal OCT images. Anisotropic pixel resolution and elongated retinal structure are the two special characteristics that make OCT retinal image registration distinguishable from other biomedical imaging modalities. Both the theoretical simulation and the experimental validation demonstrated that the normalization of $R_{L}/R_{A}$ is important for the registration of retinal OCT images with rotational movement.

We theoretically demonstrated the registration effectiveness of OCT images with an isotropic resolution better than that of the images with anisotropic resolution. First, we showed that for images with various $R_{L}/R_{A}$ values, except $R_{L}/R_{A}$ = 1, unignorable RD was observed (Figure 3). This meant that only the OCT images have an isotropic resolution; otherwise, they cannot be perfectly registered. Second, we showed that for images with anisotropic resolution and rotational movement, registration would introduce FT, which is larger in the lateral direction than in the axial direction. This phenomenon can cause significant distortion in the registration of 3D OCT/OCTA data. 

The experimental results were consistent with the theoretical simulation. The experimental validation of registration for anisotropic and isotropic resolution OCT images was performed via DR and RAR. RAR showed better performance for both the repeated OCT B-scans and the 3D OCT data. The registration performance was validated via CC (Figure 8b and Figure 9c). The compensational transformation information of the registration process was recorded, and all were consistent with the theoretical simulation.

RAR can be used to solve the registration problem caused by the anisotropic resolution of OCT images by normalizing $R_{L}/R_{A}$ before registration. The advantages of RAR can be summarized as follows.

RAR can effectively compensate for the image displacement caused by eye movement. 

The compensation for the image displacement caused by movement is the primary characteristic of an OCT registration algorithm. Compared to DR, RAR is more effective in displacement correction. The direct evidence is the CC, which was higher in the images processed with RAR for all the frames in all the data. The other evidence is provided by the mean images and en face images, which were clearer and showed more detail in RAR than in DR. 

RAR can effectively correct the image distortion caused by retinal direction differences. 

The three OCT retinal images in Figure 7a were recorded from the same eye at the same location, with different eye directions. The $R_{L}/R_{A}$ of the images was 10. The direction angular differences of the second and third images compared to the first image were 1.57 and −2.29 degrees, respectively. As Figure 7b shows, DR had poor registration results, consistent with the theoretical simulation (Figure 3), which showed that when the rotational movement is large, such as 1 or 2 degrees, the RD can be as large as several hundreds of micrometers. If we directly look at the patterns of the images, we can easily see that the patterns in these three images were very different. In other words, the second and third images were totally distorted because of retinal direction differences. That was why DR lost the capability of registering them together. However, RAR worked well for this case; the retinal direction was recovered, and the overall patterns were similar. This scenario may arise when comparing two retinal OCT images recorded at different periods, where the retinal direction changes dramatically between two acquisitions. Registration that digitally recovers the retinal direction and corrects the distortion can allow a better and more detailed comparison. This will be very helpful for longitudinal clinical monitoring and lab research. 

RAR can improve the fidelity of OCTA. 

In most quantitative OCTA studies, the blood vessel structure information is extracted from the OCTA en face image [14], such as blood vessel density [38,39,40], blood vessel tortuosity [41,42], and blood vessel caliber [39,42]. Figure 10 shows that RAR can truly represent blood vessel structures such as density, continuity, smoothness, and tortuosity, which can provide a robust precondition for OCTA quantitative feature extraction and future analysis.

RAR can preserve a large useful area of the image. 

Because of the transformation that occurred during the registration, part of the image was moved out of the image boundary, leaving a blank region in the opposite direction (Figure 10d) as a result. In the OCTA en face image, this blank region corresponded to the black regions at the top and bottom (Figure 10a). Because the blank parts contained no information, it was better to keep them as small as possible. Both lateral translation and rotation were responsible for these blank regions. The lateral translation directly moved the image out of the boundary; the rotational compensation rotated the image and moved part of the image out of the boundary. In general, the axial translational compensation can also leave a blank region. However, there is always free space at the top and bottom of OCT B-scans; thus, the useful area is not disturbed. Figure 10d shows that the en face image processed by RAR had a small black region compared to DR. The reason is that first, RAR did not introduce TF and thus had a small lateral translation; second, RAR had a much smaller rotational compensation than DR (Figure 9b), which barely generated a new blank region. Therefore, RAR can preserve a large useful area of the image.

In the simulation, the lateral FT was much larger than the axial FT (Figure 4). However, in the experiments, the compensated lateral translation was smaller than the compensated axial translation (Figure 8 and Figure 9). This is because, in the experiment, the compensated translations were composed of two parts; one was the image movement caused by the eye movement, and the other was FT. Because the eye movements along the axial direction were much larger than those in the lateral direction, the total compensated axial translation was larger than the axial translation.

One limitation of this study is that in the simulation, we only simulated cases where the foveal center was the pivot. In the real world, the pivot may not always be the foveal center; thus, the rotation change might be more complicated.

## 6. Conclusions

In this study, we proposed a new image registration strategy, RAR, for precise OCT/OCTA registration processing. We first theoretically demonstrated that without pixel resolution normalization, direct registration will introduce RD and FT, which distorts the images. To avoid RD and FT, we proposed RAR. Then, the effectiveness of RAR was demonstrated by OCT/OCTA data. This study demonstrated that RAR has better performance than DR. For OCT/OCTA image post-processing and future OCT/OCTA registration algorithm design, the normalization of pixel resolution should be seriously considered.

## Figures and Tables

**Figure 1 bioengineering-10-00313-f001:**
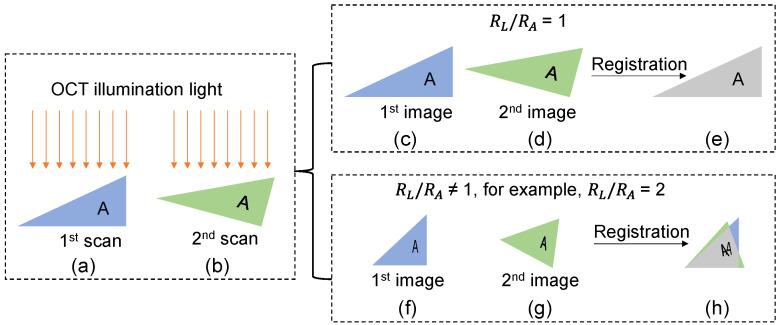
Comparative illustration of compensation performances of rotational movements in OCT scans (**a**,**b**) with isotropic (**c**–**e**) and anisotropic (**f**–**h**) resolutions. The images corresponding to the 1st and 2nd scans are shown in blue and green, respectively. The overlapping region of the two images with rotational movement corrected is indicated by the gray color in (**e**,**h**).

**Figure 2 bioengineering-10-00313-f002:**
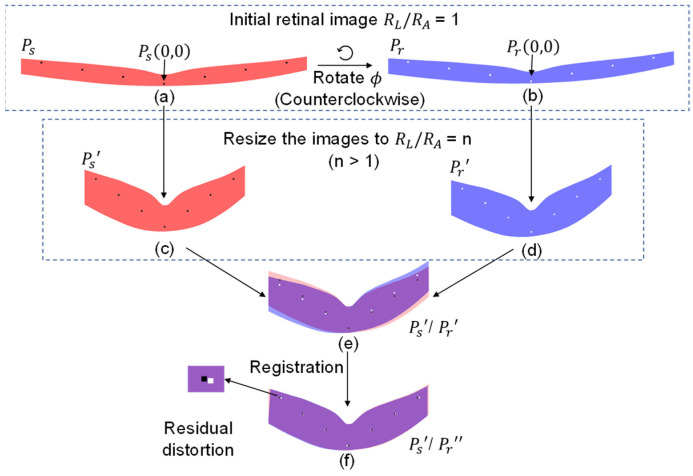
Image registration simulation of OCT retinal images with $R_{L}/R_{A}$ ≠ 1. (**a**) The initial retinal image. (**b**) The same retinal image with a rotational movement. (**c**,**d**) Image (**a**,**b**) are resized to $R_{L}/R_{A}$ = *n* (*n* > 1). (**e**) The overlap view of (**c**,**d**). (**f**) The two images in (**e**) are registered.

**Figure 3 bioengineering-10-00313-f003:**
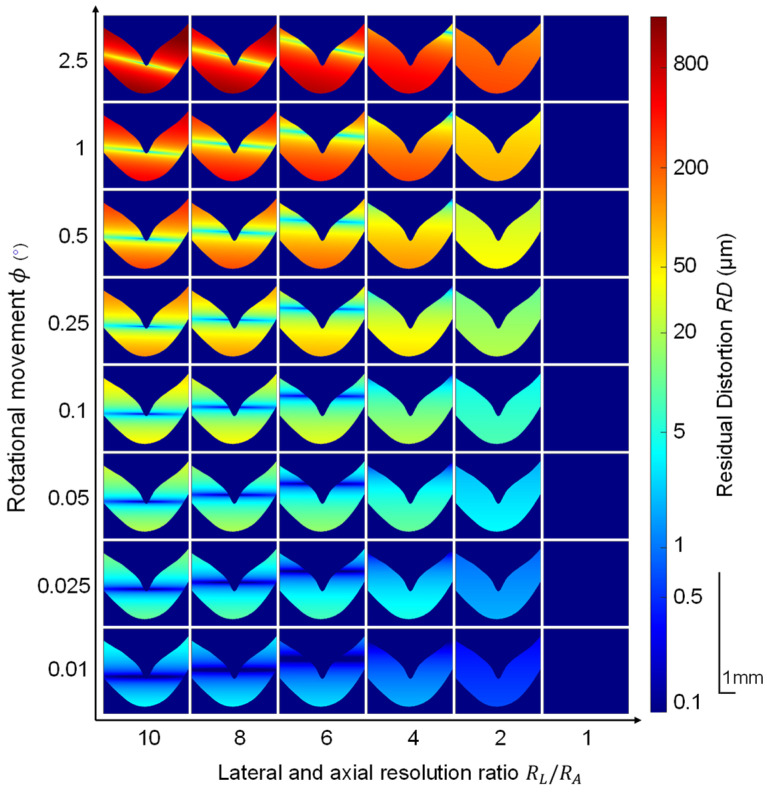
Residual distortion (RD) characterization with various $R_{L}/R_{A}$ and ϕ. The scale bar is 1 mm.

**Figure 4 bioengineering-10-00313-f004:**
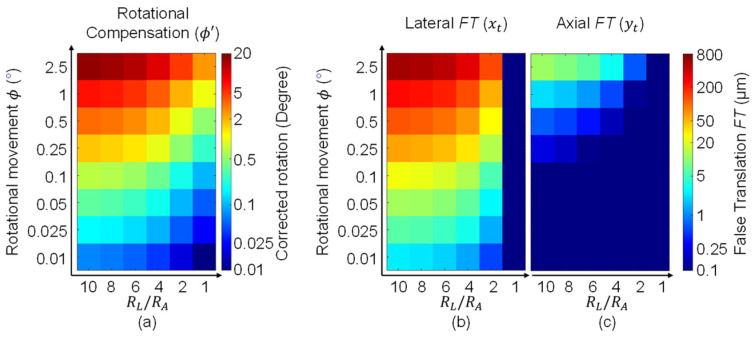
The simulation of (**a**) the rotational ($\phi^{\prime }$) and (**b**,**c**) translational ($t\left(x_{t},y_{t}\right)$) transformation by registration.

**Figure 5 bioengineering-10-00313-f005:**
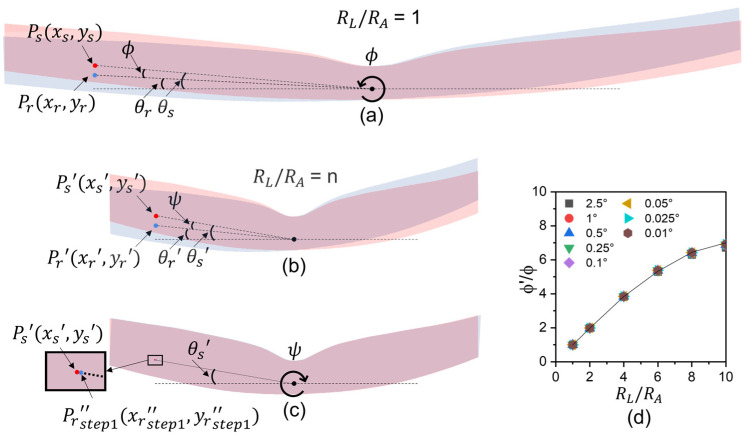
The explicit analysis of registration of OCT image with anisotropic resolution. (**a**) The original frame $P_{s}$ (red) and the rotated frame $P_{r}$ (blue) with the same point $P_{s}\left(x_{s},y_{s}\right)$ and $P_{r}\left(x_{r},y_{r}\right)$, respectively. (**b**) The images were changed to $R_{L}/R_{A}$ = n. (**c**) The rotated image was rotated back by ψ. (**d**) The ratio between $\phi^{\prime }$ and ϕ for various ϕ and $R_{L}/R_{A}$. The line shows the mean value of the points with the same $R_{L}/R_{A}$.

**Figure 6 bioengineering-10-00313-f006:**
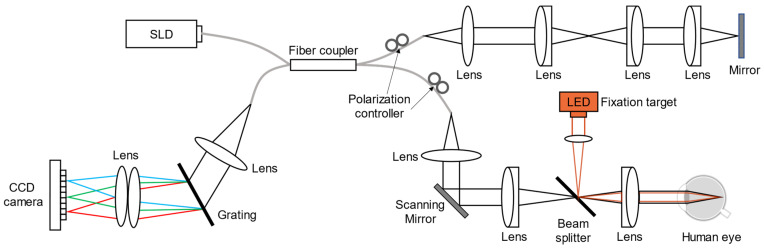
Diagram of custom-designed optical coherence tomography (OCT). SLD, superluminescent diode; LED, light-emitting diode; CCD, charge-coupled device.

**Figure 7 bioengineering-10-00313-f007:**
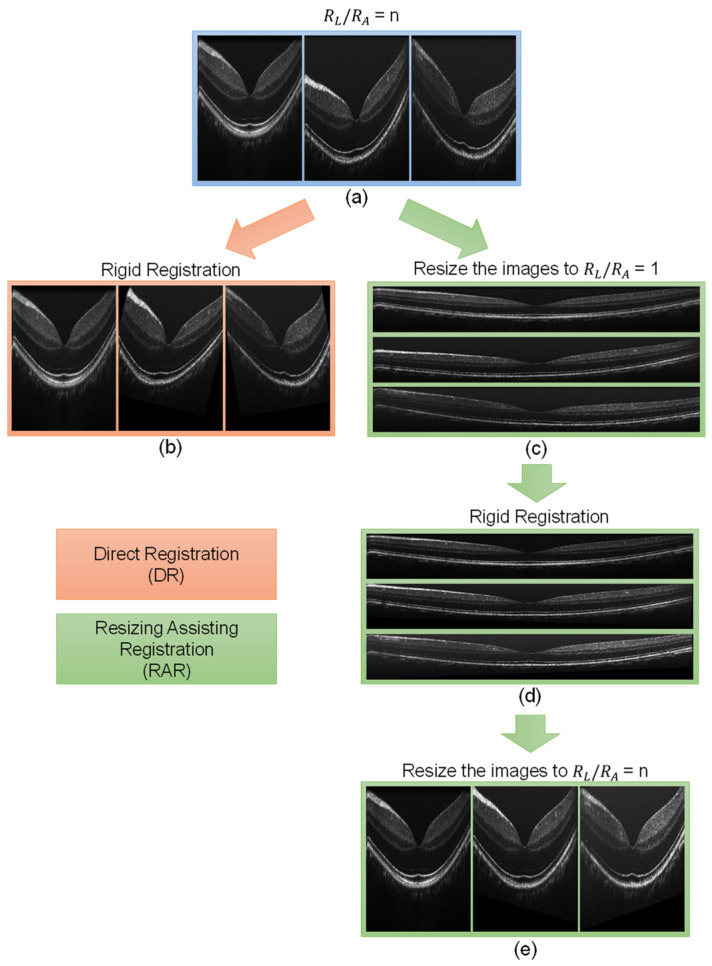
Image registration procedures of direct registration (DR) and resizing assisting registration (RAR). (**a**) Representative images with the ratio $R_{L}/R_{A}$ = *n*. (**b**) Direct registration without resizing processing. (**c**) Normalized images with the ratio $R_{L}/R_{A}$ = 1. (**d**) Registration with the normalized images in (**c**). (**e**) Resizing the image ratio $R_{L}/R_{A}$ = *n*.

**Figure 8 bioengineering-10-00313-f008:**
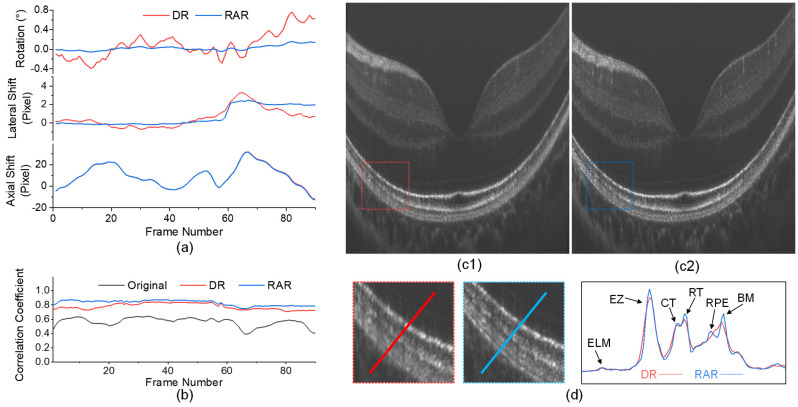
Performance comparison of direct registration (DR) and resizing assisting registration (RAR). (**a**) The rotational compensation (top), lateral translational compensation (middle), and axial translational compensation (bottom). (**b**) The 2D correlation coefficient (CC) of the original image, DR, and RAR. (**c**) The averaged B-scan images of DR (**c1**) and RAR (**c2**). Visualization 1 includes corresponding video clips before (left video) and after registration with DR (middle video) and RAR (right video). (**d**) Detailed comparison of the outer retinal layers. The images with red/blue dash line boundary are zoom-in views of the red/blue rectangles in (**c1**)/(**c2**). ELM, external limiting membrane; EZ, ellipsoid zone; CT, photoreceptor cone tip; RT, photoreceptor rod tip; RPE, retinal pigment epithelium; BM, Bruch’s membrane.

**Figure 9 bioengineering-10-00313-f009:**
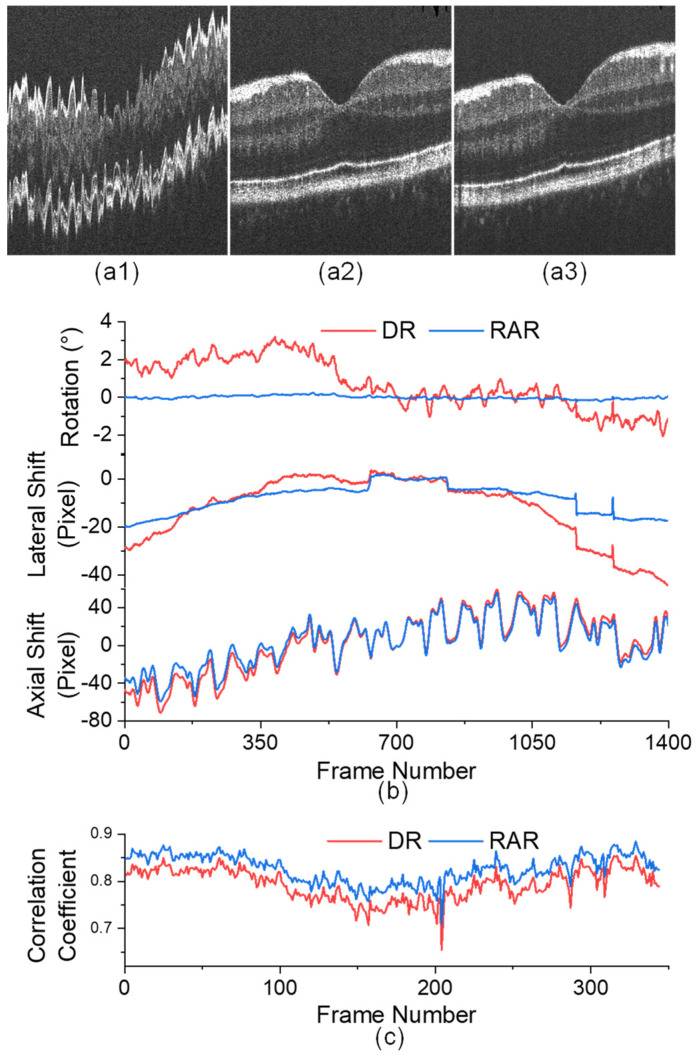
(**a**) The resliced B-scan along the slow scan direction before registration (**a1**), after DR (**a2**), and after RAR (**a3**). (**b**) The compensated rotation (top), lateral translation (middle), and axial translation (bottom). (**c**) The 2D correlation coefficient of the adjacent B-scans.

**Figure 10 bioengineering-10-00313-f010:**
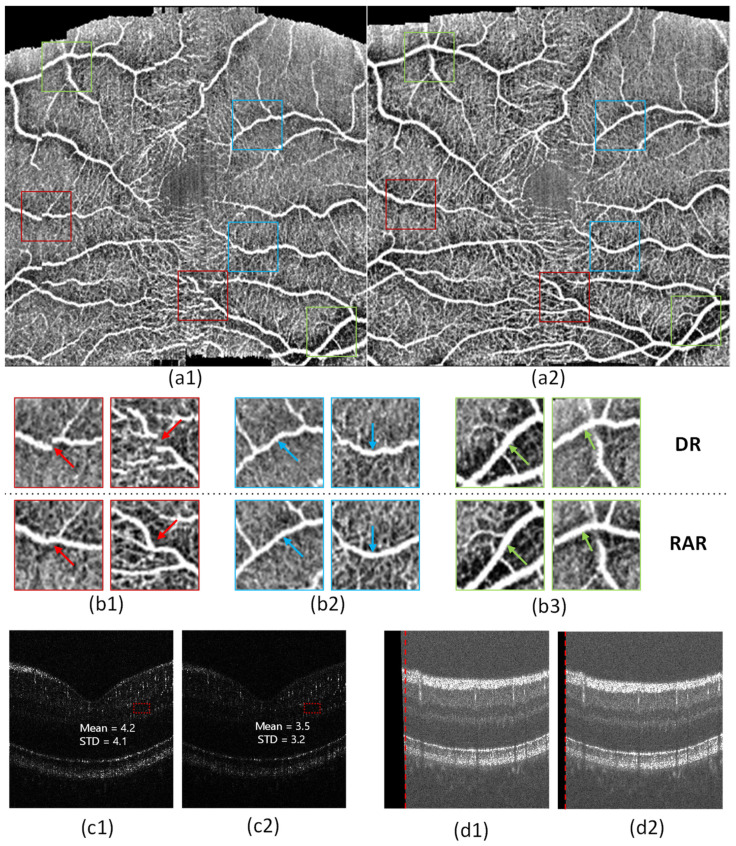
Performance comparison of direct registration (DR) and resizing assisting registration (RAR) for OCTA processing. (**a**) An en face image of OCTA with DR (**a1**) and RAR (**a2**). (**b**) A zoomed-in detailed comparison from (**a1**) (top row) and (**a2**) (bottom row). (**c**) The representative OCTA B-scans at the same location were processed by DR (**c1**) and RAR (**c2**). (**d**) The representative OCT B-scans at the same location were processed by DR (**d1**) and RAR (**d2**).

## Data Availability

The data presented in this study are available on request from the corresponding author.

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
