# Peer review of "Rotational Distortion and Compensation in Optical Coherence Tomography with Anisotropic Pixel Resolution"

_bioengineering, 2023, doi:10.3390/bioengineering10030313_

Round 1

Reviewer 1 Report

The study is very interesting and important to imagem analyses, the goal was to identify and analyze of oct image distortions due to sample (eye) movement, specially for rotation.

The notation used in figures 1 and 2 are right (a1, a2, b1 and b2....) but a think that can be improved like (a, b, c, d, e ...) it is easier to reference in the text. 

I think that de legend os figures related to RL/RA, like figure 3 and 4 should be improved, the axis as figure are no clear.

Author Response

Overall assessment:  The study is very interesting and important to image analyses, the goal was to identify and analyze of oct image distortions due to sample (eye) movement, especially for rotation.

Response: We appreciate your valuable comments and constructive suggestions.

Comment 1:  The notation used in figures 1 and 2 is right (a1, a2, b1, and b2 …) but I think that can be improved like (a, b, c, d, e ...) it is easier to reference in the text.

Response: In the revision, the notations in figure 1 and 2 were replaced with the suggested notation style (a, b, c, d, e ...).

Comment 2:  I think that the legend of figures related to RL/RA, like figures 3 and 4 should be improved, the axis of the figures is not clear.

Response: In the revision, the legend and axis of figure 3 and 4 were changed as suggested. We also changed the format of  RL/RA from regular to italic to emphasize that R_L/R_A is variable. In the previous version, figure 3 and figure 4 were more like tables instead of coordination axes. The content was listed in rows and columns. Thus we put the label of the first column and the first row in the top left corner. We hope this can address your question and confusion.

Reviewer 2 Report

The manuscript entitled "Rotational Distortion and Compensation in Optical Coherence 2 Tomography with Anisotropic Pixel Resolution" submitted by Ma et al. focuses on the OCT image registration problem induced by eye movements and anisotropic spatial resolution. Through theoretical analysis, the authors propose an image resizing assisting registration (RAR) strategy for accurate image registration and the performance is confirmed in both retinal OCT and OCTA images. The manuscript is well-written overall and the results are well demonstrated. If possible, I suggest the authors could consider adding experiment comparison on OCT model eye with artificial retinal layers which could serve as groud truth.

Author Response

Overall assessment:  The manuscript entitled "Rotational Distortion and Compensation in Optical Coherence  Tomography with Anisotropic Pixel Resolution" submitted by Ma et al. focuses on the OCT image registration problem induced by eye movements and anisotropic spatial resolution. Through theoretical analysis, the authors propose an image resizing assisting registration (RAR) strategy for accurate image registration and the performance is confirmed in both retinal OCT and OCTA images. The manuscript is well-written overall and the results are well demonstrated.

Response: We appreciate your valuable comments and constructive suggestions.

Comment 1:  If possible, I suggest the authors could consider adding experiment comparison on the OCT model eye with artificial retinal layers which could serve as ground truth.

Response: We understand and appreciate your suggestion. However, as far as we know, there is no commercially available model eye with artificial retinal layers for OCT imaging. Some of the studies fabricate such model eyes for special purposes. However, it needs complicated fabrication procedures. This study is focusing a computational method to correct residual distortion. Because the theoretical simulation properly provides the information for ground truth,  experimental comparison with the model eye seems not necessary.

Reviewer 3 Report

This manuscript theoretically and experimentally analyzes residual distortion (RD) and false translation (FT) of optical coherence tomography (OCT). An image resizing assisting registration (RAR) strategy is proposed for accurate image registration in OCT. Both theoretical analysis and the proposed RAR method are verified through quantitative comparison. I think this manuscript can be considered for its publication in Bioengineering after minor revisions. I have the following concerns about manuscript organization.

1.      This manuscript analyzes RD and FT of OCT and presents a new RAR strategy. Whether the proposed RAR method benefits from the analysis of RD and FT? If so, I suggest the authors illustrate or emphasize this point in both abstract and introduction.

2.      Title such as ‘Image resizing assisting registration for accurate OCT based on RD compensation’ will be more suitable for this manuscript.

3.      The experimental analysis in Section 5 is recommend to be placed in Section 4.

Author Response

Overall assessment:  This manuscript theoretically and experimentally analyzes residual distortion (RD) and false translation (FT) of optical coherence tomography (OCT). An image resizing assisting registration (RAR) strategy is proposed for accurate image registration in OCT. Both theoretical analysis and the proposed RAR method are verified through quantitative comparison. I think this manuscript can be considered for its publication in Bioengineering after minor revisions. I have the following concerns about manuscript organization.

Response: We appreciate your valuable comments and constructive suggestions.

Comment 1:  This manuscript analyzes RD and FT of OCT and presents a new RAR strategy. Whether the proposed RAR method benefits from the analysis of RD and FT? If so, I suggest the authors illustrate or emphasize this point in both abstract and introduction.

Response: RD and FT serve as evaluation criteria for the OCT image registration. Therefore, small RD and FT represent the image registration is accurate.   

                    In the revision, we added the sentence, “The RD and FT provide the evaluation criteria for image registration.” in both the abstract and introduction to emphasize RD and FT.

Comment 2: Title such as ‘Image resizing assisting registration for accurate OCT based on RD compensation’ will be more suitable for this manuscript.

Response: Thank you for the suggestion. As we explained in the response to your comment 1, RD is one of the evaluation criteria for accurate image registration. The RD and FT occurred because of the anisotropic axial and lateral resolution of the OCT image. Therefore, we think the “Rotational Distortion and Compensation in Optical Coherence Tomography with Anisotropic Pixel Resolution”  can be better to generally cover both RD and FT compensations.

Comment 3:  The experimental analysis in Section 5 is recommend to be placed in Section 4.

Response: In this part of the content, we summarized the advantage of RAR by comparing the results of direct registration and RAR. This part is the discussion of the results in section 4, which is why we put it in section 5 (the discussion section). 

Reviewer 4 Report

In general, your whole work is good, but couple of minor issues must be revised or added into illustration.

1.The English grammars must be checked again, such as Lines 119 and 138: “Where” should be “where”.

2. In content, if the word or abbreviation is treated as a variable, it will be shown as Italian form, such as “RL/RA” and “ RD” in lines 138-145.

3. In Figures, the denoted characters are not consistent. For instance, In Fig. 5, the illustrations are with (A), (B), (C), and (D), but they are labeled as (a), (b), (c). The final (c) should be (d). Couple similar description must be revised.

4. In Fig. 6, is there a Faraday isolator between SLD and fiber coupler or this device is mounted into SLD system? How about the ratio of beam splitter in the adjustment of light intensity, which will impact the human eye?

5. In Fig. 9, what is the error between the correlation coefficients of DR and RAR, which can be tolerated?

6. The description in article is good, but the conclusion part is too short to reflect the research integrity. 

Author Response

Overall assessment:  In general, your whole work is good, but couple of minor issues must be revised or added into illustration..

Response: We appreciate your valuable comments and constructive suggestions.

Comment 1:  The English grammars must be checked again, such as Lines 119 and 138: “Where” should be “where”.

Response: In revision, the manuscript was proofread, and “Where” in lines 119 and 138 was corrected to “where”.   

Comment 2: In content, if the word or abbreviation is treated as a variable, it will be shown as Italian forms, such as “RL/RA” and “ RD” in lines 138-145.

Response: In the revision, words or abbreviations used as variables such as “RL/RA” and “RD” were corrected as italic forms.

Comment 3:  In Figures, the denoted characters are not consistent. For instance, In Fig. 5, the illustrations are with (A), (B), (C), and (D), but they are labeled as (a), (b), (c). The final (c) should be (d). Couple similar description must be revised.

Response: In the revision, all figure denotes were corrected as (a), (b), (c), …. etc to maintain consistency, and mislabeled parts were also corrected. 

Comment 4:  In Fig. 6, is there a Faraday isolator between SLD and fiber coupler or this device is mounted into SLD system? How about the ratio of beam splitter in the adjustment of light intensity, which will impact the human eye?.

Response: The isolator is mounted in the SLD system.  The split ratio of the fiber coupler is 90:10, the 90% portion goes to the reference arm and the 10% portion goes to the sample arm to meet the laser safety regulation.

                    In the revision, this information has been added: “3.2. Imaging System and Data Acquisition”. 

Comment 5:  In Fig. 9, what is the error between the correlation coefficients of DR and RAR, which can be tolerated?

Response: The correlation coefficients are used to show image registration performance by comparing the original image with the registered image. Thus, a higher correlation coefficient represents image is registered better. In our results, the correlation coefficient of RAR is roughly 5% higher than DR through entire frames which means the RAR performance is better.

Comment 6:  The description in article is good, but the conclusion part is too short to reflect the research integrity.

Response: In the revision, the conclusion was revised to reflect the research integrity.